# Radiolabelled Aptamers for Theranostic Treatment of Cancer

**DOI:** 10.3390/ph12010002

**Published:** 2018-12-24

**Authors:** Umair Khalid, Chris Vi, Justin Henri, Joanna Macdonald, Peter Eu, Giovanni Mandarano, Sarah Shigdar

**Affiliations:** 1School of Medicine Deakin University, Geelong, Victoria 3128, Australia; khalidu@deakin.edu.au (U.K.); vchr@deakin.edu.au (C.V.); jhenri@deakin.edu.au (J.H.); J.macdonald@deakin.edu.au (J.M.); peter.eu@deakin.edu.au (P.E.); giovanni.mandarano@deakin.edu.au (G.M.); 2Peter MacCallum Cancer Centre, Melbourne, Victoria 3000, Australia; 3Centre for Molecular and Medical Research, Deakin University, Geelong, Victoria 3128, Australia

**Keywords:** aptamers, cancer, chelating agents, diagnostics, EpCAM, molecular imaging, radiolabel, targeted imaging, theranostics, therapeutics

## Abstract

Cancer has a high incidence and mortality rate worldwide, which continues to grow as millions of people are diagnosed annually. Metastatic disease caused by cancer is largely responsible for the mortality rates, thus early detection of metastatic tumours can improve prognosis. However, a large number of patients will also present with micrometastasis tumours which are often missed, as conventional medical imaging modalities are unable to detect micrometastases due to the lack of specificity and sensitivity. Recent advances in radiochemistry and the development of nucleic acid based targeting molecules, have led to the development of novel agents for use in cancer diagnostics. Monoclonal antibodies may also be used, however, they have inherent issues, such as toxicity, cost, unspecified binding and their clinical use can be controversial. Aptamers are a class of single-stranded RNA or DNA ligands with high specificity, binding affinity and selectivity for a target, which makes them promising for molecular biomarker imaging. Aptamers are presented as being a superior choice over antibodies because of high binding affinity and pH stability, amongst other factors. A number of aptamers directed to cancer cell markers (breast, lung, colon, glioblastoma, melanoma) have been radiolabelled and characterised to date. Further work is ongoing to develop these for clinical applications.

## 1. Introduction

Cancer continues to be a major cause of illness and social and economic burden. The numbers of cases increase yearly due to screening and enhanced detection methods. However, deaths arising from cancer are typically due to malignant and metastatic disease. Malignant tumours are capable of invading and spreading to surrounding tissue and to distant body sites, in a process known as metastasis, through the circulatory or lymphatic system, giving rise to secondary tumours [1]. At initial diagnosis, more than 50% of patients will have clinically detectable metastatic disease [2]. Metastatic tumours are largely responsible for cancer mortality, therefore, early tumour detection can improve prognosis [3]. Current modalities available for imaging tumour masses includes ultrasound, (X-ray) computed tomography (CT), magnetic resonance imaging (MRI) and positron-emission tomography (PET) [4] (Table 1). 

The introduction of mammography nearly 40 years ago saw a reduction in mortality from breast cancer, although this technique provides only a localised view, and whole body scans are required to detect metastatic disease. Mammography can capably identify calcific lesions, however, determining if other identified lesions (by mammography) are benign or malignant can be difficult. Despite different imaging modalities, a large number of patients will also present with micrometastases, which are often missed, as micrometastases are undetectable by conventional techniques [2,4]. 

## 2. Current Imaging Modalities and the Need for Personalized Imaging

Currently, medical imaging modalities rely on the principle of signal-to-background ratio (SBR), or tumour-to-background ratio, to create contrast within an image when the energy is attenuated by different mechanisms: soundwaves, x-ray or electromagnetism with radio-frequency waves [4,5]. Accordingly, to detect tumours, the signal generated by the tumour must be greater than that of the background signal produced by the surrounding normal tissues [4]. In order to improve SBR, contrast agents relative to each modality can be used. Contrast agents enhance imaging and identify malignant cells by distinguishing pathological cells from normal tissues on the basis of different anatomical boundaries or pathophysiology [5,13]. Contrast agents alter how the signal or energy is demonstrated by pathology when used with ultrasound, CT or MRI, to enhance the visibility when compared to adjacent tissues [5]. However, despite the use of contrast agents to enhance diagnostic medical imaging, contrast agents can be fairly non-specific to the pathology or malignant cells [4]. For example, the contrast agent in MRI, gadolinium, can be used to detect brain metastases, infectious and inflammatory processes of brain diseases, as well as characterising mass lesions in the musculoskeletal system [14]. To assess tumour biology, PET has typically incorporated radiolabelled fluorodeoxyglucose (FDG) due to the increased glycolytic rate, the Warburg Effect, exhibited by the majority of malignant tumour cells [15]. PET imaging provides information on anatomical location of tumours by exploiting the increased metabolic rate of tumours [15] and the use of radiolabeled FDG readily explains this fundamental principle. Despite advances in PET, there are challenges in molecular imaging. A major factor in molecular PET imaging is that it relies on the uptake of FDG to measure glucose metabolism by pathological cells. While pathological cells have an increased uptake of FDG, physiological uptake by normal tissues or inflammatory cells such as macrophages are highly variable, and in some instances the accumulation cannot be predicted [16]. As FDG can accumulate in normal tissues or inflammatory cells, thus decreasing availability of tracer uptake by pathological tissue, this can also lead to false positive detection and misdiagnosis [16,17]. Furthermore, FDG uptake relies on malignant cells to be metabolically active. However, some tumour cells can enter a dormant state [18]. To improve the sensitivity of diagnostic imaging, agents are being developed to target malignant cells or their products, to enable molecular imaging of a tumour [4].

Further to [^18^F]FDG, other ^18^F-labeled tracers also play a vital role in imaging and understanding tumours. [^18^F]-3′-fluoro-3′-deoxythymidine (FLT) is used for quantifying cellular proliferation [9], in particular tumour proliferation, without accumulating in inflammatory processes [19]. [^18^F]FLT can be incorporated into the patient management plan to monitor tumor biology response to treatment [19]. [^18^F]FDG and [^18^F]FLT have both been used to investigate TNBC, with mixed results. However, one study [20] evaluated response to neo-adjuvant chemotherapy in triple negative breast cancer same subject murine models with [^18^F]FDG and [^18^F]FLT and found comparable sensitivity. [^18^F]-fluoroestradiol (FES) can be used to measure estrogen binding in estrogen receptor (ER) positive breast cancer [10,21,22] and identify the aggressiveness of endometrial tumors [23]. In their study, Venema et al. showed a strong correlation between [^18^F]-FES uptake and ER expression in their patient cohort and this was confirmed with tumor biopsy and staining diagnosis. 

A number of ^18^F-labeled compounds are being tested to accurately identify tumor hypoxia [11]. This is important, as tumor hypoxia is linked with resistance to radiotherapy and chemotherapy [24]. The original and commonly used hypoxic PET tracer is [^18^F]FMISO, (fluoromisonidazole), however, it is known to be slow to accumulate at tumor sites. This was tested against second generation [^18^F]fluoroazomycin arabinoside (FAZA) and third generation [^18^F]flortanidazole (HX4) hypoxic PET tracers to determine optimal imaging time and compare their performance (tumor-to-background ratio) [11]. Uptake of [^18^F]FMISO, indicating hypoxia, correlated with glioblastoma (GBM) tumor grade; whereby Grade IV GBM demonstrated higher uptake than lesser glioblastoma grades [12]. In their study cohort, longer survival times were recorded in patients with no uptake of [^18^F]FMISO compared to patients demonstrating [^18^F]FMISO uptake.

### 2.1. Box 1

Alpha particles are emitted from an atom’s nucleus. They are relatively large subatomic particles that rapidly lose their energy to the material that they pass through. This feature makes them highly destructive to cells, and can lead to detrimental effects on all cells if delivered non-specifically. Alpha particles are known to travel up to 100 µm in human tissue [25,26] which is approximately equivalent to the dimension of 6 cells [25]. Targeted applications of alpha particle emitting nuclides are of growing interest if they can be correctly targeted to cancer sites. In this manner, they would have the advantage of eradicating cancer cells with their particle energy and minimizing radiation damage to nearby healthy cells. Clinical interest in alpha particle emitting nuclides include actinium-225 [6,25] for prostate cancer sufferers [27] and radium-223 with known skeletal uptake are used to treat certain bone cancers [6,25]. 

Compared to alpha particles, beta particles are smaller, negatively charged and travel at a faster speed. They are also emitted from an atomic nucleus, but they undergoes radioactive decay due to a high ratio of neutrons to protons. Beta particles have higher energy than alpha particles and thus have greater penetration ability. However, they can be less damaging to human cells because their ionization energy is dispersed over a greater area [28]. The most commonly used beta particle emitter used clinically is yttrium-90. It has the therapeutic advantage of treating either widespread oncology conditions such as lymphoma [29] and leukemia [30] as well as well as tumors engulfing large organs such as the liver [31,32] and pancreas [33].

Gamma rays are considered to be composed of pure energy with no mass, whereas alpha and beta particles are comprised of both mass and energy. Gamma rays have the highest energy levels of any ionizing radiation known to us in the electromagnetic spectrum and can easily pass directly through the human body. In fact, high density material such as lead or concrete is required to attenuate gamma rays [28,34]. The development of the gallium generator has introduced the widespread clinical use of gallium-68, which is now the most commonly used gamma ray emitter in Nuclear Medicine and used diagnostically with PET imaging (including PET/CT and PET/MRI). When gamma ray radiopharmaceuticles are internalized by a patient, their energy exits the patient and is detected by a gamma camera (PET camera or scintillation detector or Anger camera) [34,35].

For successful targeting of tumour sites within the human body, appropriate chelating agents are required to maintain stability between the radioactive material and the targeting modality, in vivo and in vitro [36]. Therefore, the choice of chelator used can be determined by the chemistry of both the targeting ligand and the radioactive material. Also of consideration is the behaviour or application required of the overall compound, either in vivo or in vitro [37,38]. Understanding the kinetic inertness of a chelating agent is regarded as a more reliable indicator for in vivo applications compared with its hemodynamic stability assessments [37]. In fact, chemistry experiments using a metal-exchange competition approach with biological material such as blood serum will provide directly relevant information for in vivo translation [37]. This will allow quantification of any trans-chelation that occurs from the radioactive material to serum proteins or enzymes.

## 3. Personalised Imaging Techniques—Monoclonal Antibodies

In comparison to current medical imaging modalities, molecular imaging has emerged to advance the field of medicine through the specificity, sensitivity and quantification of screening and early diagnosis of diseases [39]. Molecular imaging provides a non-invasive technique for the visualisation and characterisation of biological functions, molecular targets and cellular processes associated with in vivo [39]. The principle of molecular imaging relies on a highly specific probing molecule labelled with a radioisotope that binds to a specific receptor with high specificity and affinity, to enable external signal detection [40] (Figure 1). In order to determine the correct choice of appropriate target, typically protein expression data is required to inform selection. The use of biopsies and immunohistochemistry has provided an abundance of information in this respect. Monoclonal antibodies have long been used in diagnostic pathology to diagnose patients and find prognostic markers. Monoclonal antibodies have also been used as therapeutic agents, due to high affinity and binding to specific tumour surface antigens. For this reason, antibodies have also be repurposed and engineered to deliver radionuclide isotopes for molecular imaging [40,41,42]. With molecular imaging, antibodies are covalently linked to an imaging probe or a contrast agent to provide a signal after binding to the targeted tumour ligand, which can be detected by PET and combined with either CT or MRI to identify the anatomical location of the molecular activity [43,44]. 

Thus, radioisotopes such as gallium-68 can be attached through a bifunctional chelating approach to generate “target-specific gallium radiopharmaceuticals” for molecular imaging [45]. This approach requires a bifunctional chelator, to bind to the radioisotope with high affinity at one end of the chelating complex, to enable stability in vivo, while at the opposite end of the complex, to bind to the targeting agent [45]. To improve the application of antibodies, the therapeutic and diagnostic potential of antibodies can be combined to implement theranostic treatments—simultaneous functions of molecular imaging and targeted therapy, depending on what radioisotope is attached [42] (See Section 2.1). 

Various clinical trials are under consideration with monoclonal antibody conjugates. [^64^Cu]-NOTA-YY146 is available for lung cancer diagnostics and has shown promising results along with CD146 binding. Large numbers of CD146 antigens are present on the cell surface in glioblastoma and gastric carcinoma, suggesting other potential opportunities for this antibody conjugate [46,47,48]. ^111^In-labelled 7E11 antibodies and ^123^I-labelled anti-PSMA antibody fragment scFvD2B (PSMA) demonstrated positive results but limited use with SPECT/CT and ^111^In-labelled j591 antibodies are available for prostate cancer diagnosis via prostate specific membrane antigen (PSMA) binding [49,50]. ^111^In-j591 is available for SPECT scanning, with detection limited to antibody mass dependency. However, this antibody also demonstrated nonspecific binding to the non-prostate cancer sites, and took 5–7 days to appear in the blood stream as well as penetrate the tumor, thus making imaging quite difficult to manage [51]. In comparison [^177^Lu]-PSMA-617 and [^68^Ga]-PSMA-617 displayed better characteristics in phase 3 clinical trials without any major side effects [52]. 

Clinical studies have also been performed with [^64^Cu]-DOTA-trastuzumab as a theranostic tool in breast cancer (HER2+ and HER2−). Twenty four hours post injection, the conjugate appeared to have bound to HER2, and PET was able to produce images which demonstrated a good uptake in HER2+ cases. With copper-64 having a half-life of 13 h, the optimal imaging time window would be at this time point, when the initial blood stream concentration had reduced [53]. Another study also highlighted that [^64^Cu]-DOTA-trastuzumab conjugate saved patients from invasive biopsy procedures [54]. Zirconium-89 has also been used as a radiolabelled conjugate for metastatic breast carcinoma and exhibited promising results due to its half-life (3.3 days). This long half-life allowed for a long blood circulation time and produced better quality diagnostic images in early metastatic breast cancer [55,56,57]. It should be noted, however, that while trastuzumab binds to HER2, there was also uptake in the HER2—patients which could be due to the enhanced permeability and retention effect in solid tumours. 

Radiolabelled antibodies are also being tested to treat breast cancer and other carcinomas. Radiolabelled monoclonal antibodies used to treat breast carcinoma ideally should be designed to bind to tumor cells and be excreted from the body following their cytotoxic effects. However, the antibodies bind to their site longer than required, thus causing bone toxicity and increasing the risk of diffusion in healthy organs instead of tumor cells [58,59]. Moreover, adding DTPA or DOTA cages in radiolabelling usually diminishes radioactivity of mAbs conjugates, either because of iodination of tyrosine or addition of metal chelators [60,61]. [^177^Lu]-DTPA-Trastuzumab has been developed, but due to toxicity issues, other methods of antibody radiolabelling are under consideration [62]. A newly developed antibody PAN-622 radiolabelled with ^111^In-conjugate, [^111^In]-DTPA-PAN-622, has shown promising results in clinical trials. [^213^Bi]-DTPA-PAN-622 to HAAH has demonstrated a promising approach in a pilot therapy study of a primary tumor. Researchers are hopeful that PAN-622 mAbs will be a good theranostic tool for metastatic breast cancer [63]. [^177^Lu]-DOTA-F(ab’)2-trastuzumab was developed to overcome clearance issues, low tumor/blood and normal/tumor tissue. This conjugate shows promising stability in human serum under physiological condition and due to emission of alpha particles and gamma rays, has the potential to be used as a theranostic tool [64]. At the moment, however, only two FDA approved radiolabelled antibodies are available: ^131^I-tositumomab Bexxar^®^ (tositumomag radiolabeled with iodine-131) and Zevalin (incorporating yttrium-90) for the treatment of non-Hodgkin lymphoma and advance follicular lymphoma, [65].

Although antibodies have improved diagnostic molecular imaging and have potential to change therapeutic options, there are challenges that remain. There are several possible problems with radiolabelled antibody compounds. Firstly, tissue uptake can be reduced by covering the receptor site from free antibodies or non-specific antibodies. Secondly, cross reactivity of mAbs can cause toxicity; explained by non-specific binding of mAb radiolabel conjugate to the receptor sites. Thirdly, mAbs function could be damaged due to structural modification. Fourth, allergic reactions can occur (even after administration of a test dose) [66,67,68]. As well, with doses lower than therapeutic levels, antibodies as an imaging agent still have the potential to interfere with biological functions and illicit immunological reactions in patients [43]. Thus, an important criteria to consider when developing a novel imaging agent is for it to be biologically inert, so that it does not interfere with biological functions leading to potential safety concerns [40]. While antibody-based molecular imaging provides evidence for proof of concept that targeting cell surface or tumour biomarkers is effective, there is a need for imaging agents that have better properties.

## 4. Personlised Imaging Techniques—Peptides and Peptidomimetics

Peptides are short organic polymers. Amino acids are conjugated to each other and form a bond between multiple amino acids called peptides. Peptides may possess their own biological function or be a structural and functional part of protein molecule. Artificial peptides are also available with covalent bond between amino acid molecules [69]. Peptides are produced relatively easier than mAbs, are quicker to penetrate tumor sites and are rapidly excreted from the body [70,71]. Peptides can be used for the delivery of cytotoxic drugs and radioisotopes, as well as vaccines and hormones. Radiolabelled peptide conjugates are comprised of three parts: peptides, chelator, radioisotopes. In the late 1980’s, the first radiolabelled peptide was the Somatostatin mimic [^111^In]In DTPA-octreotide (Octreoscan), which was introduced as a gold standard diagnostic approach. Today, Octreoscan and NeoTect, ^99m^Tc-depreotide, are the only FDA approved radiotracers available commercially. Neo Tect is commonly used for lung cancer imaging and Octreoscan is used for neuroendocrine tumors [72,73,74]. [^111^In]In DTPA-octreotide has a short half-life following administration into the body and may not be able to produce desired effects. As a result, the next generation of radio-peptide therapies utilized [^90^Y]Y-DOTA-Tyr^3^-octreotide and has demonstrated more stable binding between radioisotope and chelator and demonstrated superiority in treating gastroenteropancreatic neuroendocrine tumours (GEPNETs). Also a new generation of radiolabelled peptide, [^177^Lu]]Lu DOTA,Tyr^3^-octreotate, has been introduced, improving somatostatin receptor binding affinity from six to nine fold. The use of lutetium-177 with this peptide not only provides diagnostic ability but also therapeutic effects [75]. Appropriate selection of peptide according to cell surface expression, attached to a suitable chelating agent, could develop innovative diagnostic or theranostic probes [76]. CXCR4 expression in lymph proliferative disease has been studied by using [^68^Ga]Ga-pentixafor with PET imaging and compared with [^18^F]FDG. Both tracers show uptake in lymphoma lesions. PET/MRI demonstrated the [^68^Ga]-pentixafor peptide conjugate was highly specific to tumor lesions compared to [^18^F]FDG. Superior lesion characteristics were also demonstrated in multiple myeloma patients. [^68^Ga]-pentixafor-PET/MR exhibited a high contrast between the bone marrow and the lymphoma lesions [77]. 

Common difficulties of using peptides as a drug delivery carrier are limited stability due to proteolysis by peptidases, poor transport properties through cell membranes, low oral bioavailability, rapid excretion, and poor target specificity resulting from the flexible nature of peptides. In contrast, peptidomimetics (also called peptide mimics), can be designed by the modification of an existing peptides or artificial introduction of alpha and beta amino acids in peptide structures. Peptidomimetics have fewer cross target interactions, better transport properties through biological membranes, resistance to immune responses and improved resistance to degradation by peptidases [78,79]. [^68^Ga]Ga-NODAGA-THERANOST™ is a αvβ3 integrin antagonist first used in humans for lung and breast cancer diagnosis. Two patients were imaged and treated with [^68^Ga]Ga-NODAGA-THERANOST™, the first patient (28 year old female with grade three HER2+ breast cancer and liver metastases) received 472 MBq of radiolabelled peptidomimetics conjugate and highlighted 25 lesions; however, in this same patient, [^18^F]F-FDG was only able to highlight 12 positive lesions. The second patient (61-year-old male with neuroendocrine neoplasm of the right lower lobe of the lung) received 496 MBq of [^68^Ga]Ga-NODAGA-THERANOST™ which demonstrated αvβ3 receptor expression and angiogenesis. [^68^Ga]Ga-NODAGA-THERANOST™ revealed high uptake, specificity and sensitivity in tumors and metastatic sites 60 min post injection [80]. A clinical trial is currently in progress with 120 participants with an estimated completion date in 2020 and the results may provide compelling evidence for further investigation of petidomimetics [81]. Several peptidomimetics have been developed to PSMA for both diagnostic ([^68^Ga]PSMA-11, and [^18^F]PSMA-1007 [82,83]) and therapeutic ([^177^Lu]PSMA-617 [84]) applications. These have demonstrated promising results in clinical trials though these trials have been small to date [85,86]. The results from a single arm phase II clinical trial of [^177^Lu]-PSMA-617 demonstrated high response rates though most patients did experience adverse events during the trial, predominantly grade 1 and 2 xerostomia, also known as dry mouth [87]. The results do support the need for a randomized control trial to further assess the efficacy compared to current standards of care. However, major limitations with peptidomimetics include disruption of multiple protein-protein interaction (PPI) sites and this may be the case where similar homology exists, also increasing the chances of toxicity due to off-target effects, as well as their poor serum stability [88,89,90,91].

## 5. Personalised Imaging Techniques—Aptamers

Aptamers, also known as “chemical antibodies”, are short (20–100 bases) single-stranded RNA (ssRNA) or DNA (ssDNA) oligonucleotides that bind to targets with high affinity and selectivity [92,93]. Aptamers have the ability to fold into three dimensional structures and bind to their target in a similar manner to their antibody protein counterpart via shape recognition [94]. Aptamers are generated by “Systematic Evolution of Ligands by Exponential enrichment” (SELEX) [95]. This process involves iterative rounds of incubation, isolation, elution and amplification of a randomised oligonucleotide library to a target to produce aptamers with high selectivity and specificity to the target [96,97]. Aptamers generated by SELEX demonstrates high affinity and specificity to a target, as the nucleic acids undergo iterative rounds of incubation, washing, isolation and amplification [96]. The initial library, consisting of random RNA and DNA molecules are incubated with a target of interest, biomarkers, proteins or even whole cells. Following incubation, unbound sequences are washed away to isolate and separate the bound sequences. To generate an enriched pool of sequences, the bound sequences are then eluted from the target and amplified using polymerase chain reaction (PCR) (reverse transcription PCR for RNA-based) [98]. This enriched pool of sequences then undergoes iterative rounds of selection-amplification cycles, to increase the affinity of the aptamers, as each consecutive round will decrease the heterogenicity of the pool [98]. Following analysis of binding affinities, the pool with the best affinity and specificity is then cloned and sequenced, or subjected to next generation sequencing, and further characterized to ensure specificity and sensitivity.

In comparison, aptamers hold many advantages over antibodies (Table 2). Unlike antibodies which cannot regain function after being denatured, aptamers are more stable and resistant to changes in pH and temperature, which also enables them to be easily chemically modified [99]. Antibodies require in vitro or in vivo production which can increase variation between batches, whereas this variation is reduced in aptamers as they are synthesized chemically [99]. Due to their nucleic acid composition, aptamers are generally non-immunogenic and non-toxic [92,93]. Lastly, an important advantage of aptamers is their size (5–15 kDa) in comparison to large monoclonal antibodies (approximately 150 kDa) [100]. As aptamers are much smaller than antibodies, aptamers have superior tissue penetration (greater capabilities to be internalised by tumours) [99,101]. Furthermore, the smaller size of aptamers also enables them to bind hidden epitopes which cannot be accessed by the larger antibodies [98]. Thus, given the numerous desirable properties exhibited by aptamers, the development of aptamers as molecular imaging probes is more promising than antibodies in diagnostic imaging. 

The versatility of aptamers enables the radiolabelling techniques used for targeting moieties such as antibodies, peptides or proteins, to be transferred to the aptamers with ease [98]. Coupled with the ability of aptamers to be generated and selected for against a plethora of targets with high specificity, affinity, and no toxicity, aptamers emerge as proficient radiopharmaceuticals in oncology (Table 3). The first radiolabelled aptamer was developed in 2006, TTA1 (extracellular matrix protein tenascin with fluorescent and technetium-99m tag). Aptamer uptake is dependent on the presence of human tenascin-C protein, which exists in solid tumors such as breast, colon, lung and glioblastoma. Radiolabelled aptamers have been observed in glioblastoma (U251) and breast cancer (MDA-MB-435) xenograft models on planar scintigraphy. Intravenous administration of [^99m^Tc]-TTA1 aptamer was also observed in mice at different time intervals in biodistribution and imaging studies, with signal first observed at 10 min and at 3 h complete tumor diffusion was observed as well as renal and hepatic clearance. Rapid tumor uptake, high renal clearance rate to target specific binding sites and ideal conjugation of aptamer with tumors raised the importance of aptamers to be used as a theranostic tool [103]. [^99^Tc]-TTA1 in murine studies highlighted the importance of aptamer as diagnostic probes. In another study that investigated the use of technetium-99m conjugated to an aptamer that binds to the MUC1 receptor, which is highly expressed in various cancers such as lung, breast and prostate, a good tumour-blood ratio was observed at 16 h [99,104]. However, further work is required to turn this into an efficient diagnostic probe.

Conjugation of aptamers with a chelating agent is considered an important chemistry step, as it provides in vitro and in vivo stability. There have been a number of aptamers tested with DOTA, NOTA, MAG_3_ and DTPA chelators. ^99m^Tc-MAG_3_ (Technetium-99m) [^99m^Tc] mertiatide (MAG_3_) has been used for analysing kidney function with nuclear medicine imaging. [^99m^Tc]-MAG_3_ forms a stable chelate and can act as a bifunctional chelator, in biological radiolabelling but due to complicated conjugate properties and radiolabelling chemistry, yield has always appeared low in comparison to the expected yield. Researchers highlight the cause of low yield, which was not only incomplete chemistry conjugation and purification of radiolabelled aptamer conjugates, but also due to labelling of impurities. Preliminary purification before labelling and use of rhenium-188 has been suggested to resolve this issue [38]. Thus a purified form of radiolabelling could achieve better results. In this study researchers used [^99m^Tc]-MAG_3_-F3B aptamer with high radiochemical purity and yield against hMMP-9 in melanoma. Quantitative biodistribution was assessed at different intervals, with 1.8% ID/g observed in the tumour at 1 h, which was maximum uptake of the conjugate. However, a large amount of digestive tract accumulation and retention was observed (18% ID/g at 1 h), suggested to be due to lipophilic nature of the radiolabelled compound. Therefore, [^99m^Tc]-MAG_3_-aptamer conjugate would be difficult to use for determining digestive and abdominal region [105]. It is likely that this accumulation would prevent detection of metastases in this area, as well as in others due to the decreased bioavailability and thus precludes the use of this chelator for future studies.

[^64^Cu]-DOTA-AS1411, [^64^Cu]-NOTA-Bn-AS1411 and [^64^Cu]-CB-TE2A-AS1411 were developed for lung cancer diagnosis and are currently in animal trials. Non-small lung cancer cells were implanted in right and left thighs of BALB/c nu/nu female mice and after two weeks, when tumor growth was at 0.5–1.0 cm^3^, PET/CT were performed. Approximately, 7.4 MBq (200 uCi) were injected via tail vein. Anesthetized mice were then scanned at 1, 3, 6 and 24 h after injection. The percentage injected dose per gram were used to calculate radioactivity, however, chelator-AS1411 yield were 86% +/− 7% for DOTA, DOTA-Bn, NOTA-Bn and 78% +/− 10% for CB-TE2A. The radiolabeling yield and specific activities were 65 ± 15, 57 ± 19, 15 ± 6, 75 ± 24%, and 45 ± 26, 48 ± 16, 2 ± 1, 9 ± 4 mCi/μmol for [^64^Cu]-DOTA-AS1411, [^64^Cu]-CB-TE2A-AS1411, [^64^Cu]-DOTA-Bn-AS1411 and [^64^Cu]-NOTA-Bn-AS1411, respectively. Percentage uptake was affected by the chelator used. Copper-64 [^64^Cu]-DOTA-AS1411 uptake increases dramatically in the first 6 h and at the 12th h reached a stable phase and commenced clearing from the body in 48 h. Copper-64 [^64^Cu]-CB-TE2A-AS1411 demonstrated peak reach at blood plasma level concentration at 3 h with a lag phase at 9 h and no significant clearance in 48 h. This data demonstrated that while CB-TE2A may have better properties for radiolabelling yield, the fact that it was not being cleared from the body at 48 h would lead to a low tumour-background ratio and thus, missed metastases.

In a different study, Li et al. radiolabelled AS1411 aptamer with ^64^Cu, and further explored the effects of four different bifunctional chelators; DOTA, CB-TE2A, DOTA-Bn and NOTA-Bn, on the pharmacokinetics of the aptamers in targeting lung cancer. In vivo results demonstrated high binding affinity to targeted cells and sufficient tumour uptake in H460 tumour bearing mice models, with ^64^Cu-CB-TE2A chelated being the most efficient [108]. In comparison to both, [^64^Cu]-NOTA-Bn-AS1411 showed least uptake and proved an effective radiolabeling chelator. In further in vivo studies [^64^Cu]-CB-TE2A-AS1411 showed better pharmacokinetic results and low liver uptake in comparison to [^64^Cu]-DOTA-AS1411. [^64^Cu]-DOTA-AS1411 could not detect lung tumor with micro PET imaging. In addition to that, in vivo biodistribution were performed after each micro PET/CT scan at 24 h post injection. All major organs such as spleen, liver, kidney, heart and lungs showed reduction in accumulated radioactivity over 90% in comparison to [^64^Cu]-DOTA-AS1411. However, tumor uptake reduced by one half (1.17 ± 0.04% ID/g for [^64^Cu]-DOTA-AS1411 vs 0.56 ± 0.37% ID/g for [^64^Cu]-CB-TE2A-AS1411). [^64^Cu]-CB-TE2A-AS1411 express excellent in vivo kinetic stability with faster clearance and higher tumor-to-background ratio at 24 h post injection. Tumor-to-blood and tumor-to-muscle ratio for [^64^Cu]-CB-TE2A-AS1411 were higher than [^64^Cu]-DOTA-AS1411 [108]. The anomaly being that even though high levels of chelation were achieved, tumor uptake may not be at the level desired. These studies indicate that the choice of chelator is important when considering the development of radioactive ligands for both targeted imaging and therapeutics.

[^18^F]fluorobenzyl azide-Heraptamer-1 and [^18^F]fluorobenzyl azide-Heraptamer-2 conjugates have been developed for HER2 cancer and in vitro studies were performed with HER2 extracellular domain, using the HER2 positive SKOV3 cell line and the HER2 negative MDA-MB-231 cell line. Different concentration of heraptamers were prepared, incubated with HER2+ and HER2− cell lines, then analysed by flow cytometry. Both aptamers show strong fluorescent signals with the positive cell line and no binding with the negative cell line. Results were obtained at low nanomolar concentration suggesting specificity and strong binding affinity of heraptamers. Also, no cytotoxicity was observed with SKOV3 cell line following 2 days of treatment. Following in vitro studies, direct radiolabelling of aptamers were used in PET imaging of mice. The conjugate was injected intravenously in a xenograft mouse model followed by PET scan at 30 min of post injection. Results varied from the in vitro studies with SKOV3 cell line. [^18^F]-heraptamers rapidly cleared from the renal route and demonstrated high uptake in bladder and kidneys. The gallbladder also showed high uptake due to metabolism of aptamer. Further studies performed in tumor bearing mice with ovarian cancer (SKOV3) demonstrated heraptamer-1 at 15 min post injection (1.04 ± 0.18% ID/g) and heraptamer-2 (0.67 ± 0.10% ID/g) uptake; at 1 h of post injection heraptamer-1 decreased (0.52 ± 0.04% ID/g) and heraptamer-2 (0.41 ± 0.14% ID/g), representing reduced uptake at tumor sites. At 1.5 h, organs were collected and analyzed for biodistribution of radiolabelled aptamers. High background signal properties (tumor-to-blood ratio) was observed, which represented an ideal approach for diagnostic purposes and resolved all uptake and degradation issues. Due to successful pre-clinical trials, this gives hope that these aptamers may provide future diagnostic purposes [109].

Conjugation of different molecules to aptamers enables their functionality to be modified accordingly. For example, Wang and Farokhzad [99] conjugated the A10 aptamer, targeted against Prostate-specific membrane antigen (PSMA) on prostate cancer cells, to superparamagnetic iron oxide nanoparticles (SPIONs). While the A10-SPION conjugate demonstrated high sensitivity and specificity for MRI detection at the targeted location, the study also showed that the resulting construct was capable of carrying and delivering a chemotherapeutic drug, doxorubicin, to the PMSA on prostate cancer cells [99]. Furthermore, recent studies with [^177^Lu]-PSMA labelled peptides, demonstrated the theranostic capabilities of the lutetium-177 due to its ability to emit beta and gamma radiation simultaneously [110]. The beta-radiation is capable of destroying small tumours, due to maximal tissue penetration of 2 mm, while gamma-radiation enables PET imaging of the tumour [110]. With the ease of conjugating aptamers, radiolabelling it with lutetium-177 radioisotope offers promising and novel theranostic capabilities.

The epithelial cell adhesion molecule, EpCAM, was one of the first cell surface markers to be associated with cancer [102] and has been associated with a poor prognosis in a number of different cancer subtypes [111,112]. EpCAM is a type I transmembrane glycoprotein highly expressed in epithelial cancers. In normal cells, EpCAM is localized to the basolateral membrane, but during cancer progression, the expression pattern changes to an intense uniform over-expression [113]. As EpCAM is over-expressed in epithelial cancers, it was thought to be a good target for directed therapies. However, to date, this has not been realized for a number of reasons. While it is over-expressed in cancer cells, it is also expressed in normal tissues. The first targeted therapies to EpCAM had a high affinity for their target and were unfortunately poorly tolerated by patients. This was theorized to be because the high affinity meant that the antibodies were binding to all EpCAM expressing cells, not just the highly expressing cancer cells. The next antibody that was generated therefore was engineered with a lower binding affinity to EpCAM. This was texted in patients with metastatic breast cancer and did show some efficacy. However, this efficacy was demonstrated only in patients that had a high expression of EpCAM on the cancer cells surface [114]. Given that there are few markers to utilize for targeted therapeutics for triple negative breast cancer and that EpCAM is present in close to 64% of these patients’ tumours [115], it may represent a viable target for targeted medical imaging. A recent review article highlighted the issue of using antibodies against EpCAM but has highlighted the potential for using aptamers as targeted therapeutic delivery agents [102]. As detailed below, these aptamers are in the early stage of clinical development but have shown promise so far. As well, given the increased interest in radiolabelling aptamers as a viable alternative to antibodies, these may increase our diagnostic and theranostic capabilities for cancer patients, including patients diagnosed with triple negative breast cancer given there are no available targeted therapies to date.

A study by Xiang, et al. [101] demonstrated the superiority of EpCAM aptamers compared to monoclonal EpCAM antibodies in molecular imaging in vivo using xenograft mouse models. In this study, the EpCAM-targeting aptamer was conjugated to a DY647 fluorophore and injected intravenously into the tumour bearing mice. The aptamers demonstrated greater tumour penetration, accumulation and retention for molecular imaging, compared to antibodies, as they can increase the signal strength to be detected by the imaging scanners [101,116]. This aptamer, generated by Shigdar and colleagues, is a highly specific RNA aptamer used to target EpCAM. The specificity and binding affinity of the RNA aptamer was assessed against live human cancer cell lines expressing EpCAM and those that did not express EpCAM (breast, colorectal and gastric cancers) and analysed using flow cytometry. These modified RNA aptamers also demonstrated good bioavailability though DNA aptamers are less costly to work with [92]. Alshaer et al. developed a DNA aptamer, Ep1, and and demonstrated the generated Ep1 aptamer’s ability to bind specifically and selectively to the EpCAM protein. The binding affinity and selectivity of the Ep1 aptamer was analysed using an EpCAM-positive gastric cancer cell line, KATO III, and comparing it to an EpCAM-negative mouse fibroblast, NIH/3T3 cells. The study showed greater aptamer-EpCAM interaction for the Ep1 and KATO III cell line, when analysed by flow cytometry and immunofluorescent assays [117]. Additionally, Song, et al. [118] have previously developed a 48 nucleotide DNA aptamer, SYL3C, which consists of three hairpin loops, targeting the EpCAM biomarker and demonstrated strong binding affinity and selectivity, using the MDA-MB-231 breast cancer cell line and KATO III gastric cancer cell line. Interestingly, Song et al. were able to also demonstrate the SYL3C aptamer’s specificity by targeting the EpCAM biomarker on cancer cells in a mixed cell media. To further enhance sensitivity and specificity, the SYL3C aptamer was truncated by Macdonald, et al. [119], into three separate hairpins, EpA (nucleotides 5–21), EpB (nucleotides 22–37) and EpC (nucleotides 39–46). All three truncated aptamers demonstrated greater sensitivity in binding to EpCAM positive cell lines, HT29 (high level of EpCAM expression), and HEY (low level of EpCAM expression), when compared to the full length SYL3C aptamer. This study found that the truncation may have affected the 3-D conformational structure of both EpB and EpC, as they were able to bind to a high EpCAM targets (HT29) with high affinity but could not completely interact with lower levels of EpCAM (HEY), when compared to EpA which had greater affinity with the HEY cell line. The aptamers generated are highly specific to targeting the EpCAM biomarker on cancer cells, as a result, this suggests the promising applications of utilising aptamers as in vivo molecular imaging agents.

While much has been done to both diagnose and treat systemic tumours, one of the areas of research lacking in any success to date has been the diagnosis and treatment of brain metastases. The numbers of brain metastases are increasing on a yearly basis though, because of the blood brain barrier (BBB), they are difficult to treat. Additionally, there are few techniques available to adequately diagnose small brain metastases. Both [^18^F]FDG PET and MRI have been used to diagnose brain metastases with varying accuracy. It has been demonstrated that PET detection is highly dependent on size, with metastases smaller than 5 mm being unlikely to be detected. MRI can detect lesions smaller than 5 mm but can also present false positives that require additional imaging to confirm. MRI may also miss the micro-metastases present in patients if they are in good neurological condition [120]. As well, some patients cannot undergo MRI, due to implants or metal foreign bodies, leaving these patients with a much poorer prognosis due to reliance on PET or CT to pick up their brain metastases. Recently, a paper was published suggesting a bifunctional aptamer can be used as a delivery vehicle to not only cross into the brain across the BBB, but also only target metastatic tumours derived from epithelial cancers once there. This aptamer combination firstly targets the transferrin receptor on the BBB to use receptor mediated transcytosis to non-invasively move across the BBB into the brain. The second arm of this aptamer uses the previously described EpA aptamer to then specifically target the tumour cells. When injected in a healthy animal model, this aptamer was able to accumulate in the brain within 10 min [119]. Further ongoing work with this aptamer is assessing its ability to act as a molecular imaging probe, which if successful, could pave the way for other targeted imaging agents to detect micro-metastases in the brain. The majority of these studies have investigated the specificity of these EpCAM aptamers to several different cell lines, representing a number of different cancer types, and suggest potential future directions for developing targeted therapeutics. There has also been some in vivo studies that suggest the efficacy and tolerability of these aptamers which make them well placed for further development. 

Practical applications of aptamers in vivo demonstrated some challenges such as susceptibility to degradation by nucleases present in human serum and the rapid excretion by renal filtration [93]. In human serum, due to the presence of nucleases, both unmodified RNA and DNA aptamers are susceptible to degradation, with RNA being more sensitive due to the 2′Hydroxyl group at the ribose location [96,98]. Fortunately, the short half-life of aptamers in human serum can be extended, as aptamers can easily be modified to increase resistance to exo- and endo-nuclease degradation [96]. The modified nucleotides can be introduced pre or post selection process via SELEX; with pre-selection having limited options due to incompatibility of modified nucleotides with the enzymatic steps of SELEX [98]. In comparison, post-selection modifications are easier and cheaper, but can negatively affect the binding properties and functionality of the aptamer [96,98]. 

A common method to prevent aptamer degradation by exo-nucleases, is to cap the 3′- and/or 5′- ends of the nucleic acid strands [121]. However, to increase stability to endonuclease-mediated degradation, the inter-nucleotide linkage can be modified with a phosphate backbone being substituted with sulfur to create a phosphorothioate linkage [122,123]. In the case of RNA based aptamers, the serum half-life and RNase resistance can be increased by modifying the highly nuclease susceptible 2′-hydroxyl group of the ribose sugar [96,98]. These modifications involve replacing the 2′-hydroxyl group with either fluoro (F), alkyl, amino (NH_2_) or thio group [124].

Since aptamers are relatively small molecules (5–15 kDa), they are easily excreted by the kidneys [93]. Thus, renal filtration of aptamers leads to challenges in practical application due to the decreased circulation in the bloodstream [93]. A current solution to delay the excretion of aptamers from the bloodstream and improve aptamer pharmacokinetics, is conjugation with cholesterol or polyethylene glycol, termed PEGylation, to increase serum retention time of low molecular weight substances [93,122]. 

## 6. Conclusions

Although cancer is a leading cause of mortality globally, early diagnosis and detection can improve treatment outcomes due to early surgical, curative intervention. The challenges lie in present medical imaging and diagnostic techniques in oncology. While current medical imaging modalities can identify tumour masses, they are unable to specifically detect micrometastases before their angiogenesis stage, due to the minimum number of cells required for detection. PET molecular imaging has been able to improve detection of malignant cells, however, the typical use of FDG for pathological cells is non-specific for a disease, as normal tissues can uptake FDG which increases the background signal relative to the tumour. A number of different targeting ligands have been investigated in pre-clinical studies, though more work has been completed investigating antibodies and aptamers (Table 4). Antibody and peptide-based imaging has significantly improved molecular imaging, as it provides evidence to support the concept of targeting cancer surface molecules or biomarkers for highly specific and sensitive imaging of a pathology. However, both can interact with biological functions in vivo which can lead to adverse reactions, due to immunogenic responses. Therefore, with the advancement of nucleic acid chemistry, aptamers can be generated and modified to serve as molecular imaging agents with great specificity to a malignant cancer cell. Similar to antibodies, aptamers demonstrate great capacity to selectively and specifically bind to a cancer biomarker with high affinity to enable targeted imaging. In a similar process to radiolabelling antibodies, aptamers can be radiolabelled with radioisotopes such as gallium-68, to enable signal detection by PET/MRI scanners. In contrast to antibodies, aptamers offer numerous advantages: smaller size, greater stability and non-immunogenic nature in vivo. Limited work has been completed in this area but combining nuclear medicine techniques and radiolabeled aptamers will provide new opportunities to treat cancer at a cellular level or at a metastatic stage and provide opportunities for early diagnosis. Aptamers are an emerging frontier in medical molecular technology for cancer diagnostics and therapeutic applications. With their shared advantages and limited comparative disadvantages as compared to antibodies, it is hoped that aptamers will provide a better diagnostic and therapeutic option for patients in the future. 

## Figures and Tables

**Figure 1 pharmaceuticals-12-00002-f001:**
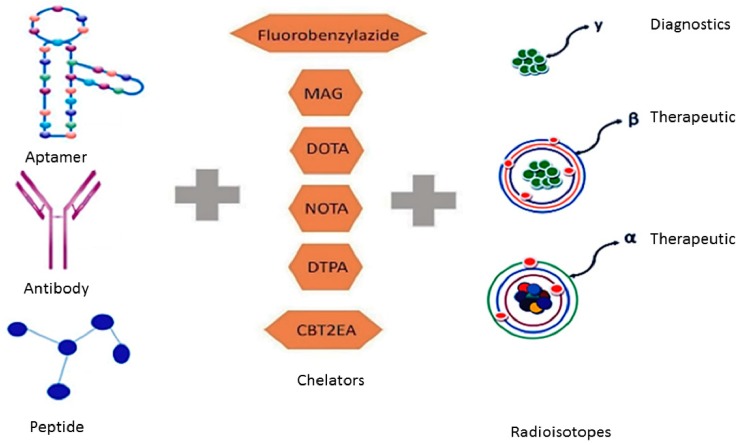
Targeting ligands are conjugated to a chelator which can bind a radioisotope that can be used for diagnostics or therapeutics.

**Table 1 pharmaceuticals-12-00002-t001:** The different methods of malignant tumour cell detection by current medical imaging modalities and its advantages and disadvantages. Source adapted from [4,5,6,7,8,9,10,11,12].

Medical Imaging Tool	Method of Detection	Advantages	Disadvantages
**Ultrasound**	Soundwaves (1 to 10 MHz) to visualise soft tissue Soundwaves are scattered depending on tissue density (echogenicity) Hypo-echoic (darker pathology image due to tumour being of lighter density than surrounding tissue) Hyper-echoic (brighter pathology image due to tumour being of greater density than surrounding tissue)	Non-ionising radiation High resolution Cross-sectional anatomy representation Readily available and accessible Real time information	Limited depth (approximately 10 cm) Operator dependent Unable to detect capillary network
**X-Ray CT**	X-ray beams through the body Measures attenuation of x-ray due to tissue density	Fast acquisition times Greater sensitivity 3-d image reconstruction	Ionising radiation Requires contrast media with high atomic number
**MRI**	Magnetic fields and radiofrequency signals Align and rotate the magnetic spin of protons Measures the time taken for spin of proton to return to normal state—relaxation time Hyper-intense Hypo-intense	Non-ionising radiation Superior soft tissue definition Multiplanar reformation Superior spatial resolution	Not all patients can enter magnetic environment (patients with pacemakers, aneurysm clips) Contrast media required, which can lead to adverse event Expensive
**PET**	Visualization, characterisation and quantification of metabolic processes at cellular and sub cellular level in body. Positron emitting radioisotopes provide pairs of gamma rays (180 degrees to each other) with 511 KeV diagnostic energy, identified with gamma camera. Examples for diagnostic purposes include fluorine-18, gallium-68, copper-64. An example of a theranostic radioisotope is lutetium-177. Radiolabelling of a variety of markers and molecules: FDG (glucose metabolism), FLT (quantification of cell proliferation), FES (measure regional estrogen binding), MISO (evaluate tumour hypoxia).	Ideal tool for early diagnosis and targeted imaging. Optimize gene and drug therapy. Simultaneous monitoring of molecular events in body.	Radioisotope used depends on biochemistry application. Expensive diagnostic method.
**SPECT**	Direct imaging of photon energy (gamma ray). Utilizes single photons emitted by gamma-emitting radioisotpes such as technetium-99m, indium-111 and iodine-123. Spatial resolution clinical aspect 8–12 mm.	Explains the function of, and blood flow to, organs.	Longer biological half-life. Increased probability of detecting secondary tumors.

**Table 2 pharmaceuticals-12-00002-t002:** Comparison of antibodies versus aptamers (adapted from [102]).

Property	Antibodies	Aptamers
Reproducibility	Batch-to-batch variation	No batch-to-batch variation
Cost	Can be expensive to generate	Generally cheaper to generate
Generation time	Lengthy	Can be rapid
Specificity	Can be cross reactive	High specificity
Tumour penetration	Slow	Rapid
Stability	Sensitive to pH and temperature	Stabile in range of pHs and temperatures

**Table 3 pharmaceuticals-12-00002-t003:** Examples of radiolabelled aptamers.

Radioisotopes	Chelator	Aptamer	Target	Application	References
^99m^Tc technetium-99m	N/A	TTA1	MUC1 receptor	Breast, lung colon and glioblastoma	[103]
^99m^Tc technetium-99m	MAG	F3B-aptamer	hMMP-9	Malignant melanoma	[105]
^99m^Tc technetium-99m	DOTA	F3B-aptamer	hMMP-9	Malignant melanoma	[106]
^111^In indium-111	DOTA	F3B-aptamer	hMMP-9	Malignant melanoma	[105]
^6^^4^Cu copper-64	DOTA	AS1411	Large nucleolin complex	Lung cancer	[107]
^64^Cu copper-64	NOTA-Bn	AS1411	Large nucleolin complex	Lung cancer	[108]
^64^Cu copper-64	CB-TE2A	AS1411	Large nucleolin complex	Lung cancer	[108]
^18^F fluorine-18	Fluorobenzylazide	Heraptamer-1	HER2 over expression	HER2+ cancer	[109]
^18^F fluorine-18	Fluorobenzylazide	Heraptamer-2	HER2 over expression	HER2+ cancer	[109]

**Table 4 pharmaceuticals-12-00002-t004:** Examples of targeting ligands that have a potential to bind with radioisotopes for diagnostic and therapeutic purposes.

Ligands	Target Cells	Advantages	Disadvantages	References
Albumin	Breast cancer cells	Longer half-life, low intrinsic activity, stable to adverse pH and temperature	Long circulatory half-life, Accumulate in healthy tissues.	[125,126]
Antibodies	Multiple targets	Targeted delivery of drug	Batch to batch variations, cross reactivity	[127,128]
Aptamers	Multiple targets	Non immunogenic, small in size	Rapid renal excretion	[93]
Fibrinogen	Tumor vasculature	Covalent linkage system to provide control drug release	Not highly specific to the receptor sites.	[129]
Folate	Leukemia cells	Receptor targeted delivery with low level of toxicity in normal tissue	Non-specific binding, pH sensitive linkage	[130]
Hyaluronic acid	CD44+ melanoma cells	Sustain release of drug	Chemically un-stable in body.	[131]
Peptides	Bombesin, gastrin/cholecystokinin-2	Small in size, low toxicity and highly specific	Higher proteolytic in-stability	[91,132]

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
