# Peer review of "Radiolabelled Aptamers for Theranostic Treatment of Cancer"

_pharmaceuticals, 2018, doi:10.3390/ph12010002_

Round 1

Reviewer 1 Report

The manuscript “Radiolabelled targeting ligands…“ addresses the advantages and disadvantages of using either antibodies or aptamers as targeting moieties for theranostic applications in Cancer. The topic is timely and informative for a rapidly evolving field. The manuscript is fluid and coherently organized.

We suggest:

1) to include a table showing the advantages and disadvantages of antibodies versus aptamers as ligands for theranostic use.

2) To include a scheme illustrating the more effective manners that have been adopted to link a nucleic acid aptamer to the diverse imaging probes

3) Since other tumors are described in the Review besides breast cancer, it would be better to delete “breast” from key words, and change accordingly the Introduction

Author Response

Thank you to the reviewer for their comments. Please see below our responses:

We suggest:

1) to include a table showing the advantages and disadvantages of antibodies versus aptamers as ligands for theranostic use.

We have added in a table of the advantages and disadvantages of antibodies versus aptamers

2) To include a scheme illustrating the more effective manners that have been adopted to link a nucleic acid aptamer to the diverse imaging probes

Following reviewers comments, we have added a schematic that demonstrates how targeting ligands attach to chelators and which radioisotopes are effective for different purposes

3) Since other tumors are described in the Review besides breast cancer, it would be better to delete “breast” from key words, and change accordingly the Introduction

We have deleted breast from the key words.

Reviewer 2 Report

The review describes the use of nucleic acids based radiopharmaceuticals for as a novel agents for cancer theranostics. Thus, I would recommend including the word cancer to the title, or adding the information on other applications of radiolabeled aptamers.

The article is quite interesting and starts with the description of current cancer imaging techniques, basic principles of radioactivity as a diagnostic tool, and aptamer nature and selection. The most interesting part includes the description of aptamers based radiopharmaceuticals.

I have some suggestions how this manuscript could be improved for the better understanding.

1. Despite the words: molecular imaging; radiolabel; targeted imaging are listed as a key words, the paper does not contain figures. The reader would expect to find in this paper how all this imaging looks, ideally in comparison with other non-radioactive methods. Adding the figure with the principal sheme of radiolabeling of the aptamer and targeting will make it easier for the reader to understand and attract people. Most of the readers firstly look at the pictures and then read.

2. It is a good idea to represent the table 1 with advantages and disadvantages, please include also PET and SPECT to this table.

3. Abstract should be more specific. The number and some specifications and applications of radiolabeled aptamers should be included into the abstract.

4. Please make the table with all aptamer based radiopharmaceuticals.

Which should contain: radioisotope, chelator, aptamer, target, application, reference (may be something else)

Why does the introduction starts with Breast cancer. And only this cancer type is described? Why not lung or brain. I think it should be more general.

Line 163-165: Clinical studies have also been performed with 64Cu-DOTA-trastuzumab as a theranostic tool in breast cancer (HER2+ AND HER2-) After 24 hours post injection, the conjugate appeared to have

bound to HER2, and PET was able to produce images which demonstrated a good uptake in HER2+ cases.

No information on HER2-

Line 164 (HER2+ AND HER2-) – and should be in lowercase

Lines 200. Will be good to have picture or table with criteria for developing a novel imaging agents.

Also picture or table with all possible targeting ligands suitable for radiolabeling with advantages or disadvantages

Line 207 aptamers could be up to 100 nucleotides

Line 223 you described selection but looks like as the last  pool will be sequenced, it is better to write that after analyzing binding abilities the pool with the best affinity and selectivity will be sequenced.

Line 223 Now most of researchers prefer NGS sequencing, but not cloning with further sequencing, as it was before.

Author Response

We thank the reviewer for their comments. Please below for our responses:

The review describes the use of nucleic acids based radiopharmaceuticals for as a novel agents for cancer theranostics. Thus, I would recommend including the word cancer to the title, or adding the information on other applications of radiolabeled aptamers.

We have amended the title

The article is quite interesting and starts with the description of current cancer imaging techniques, basic principles of radioactivity as a diagnostic tool, and aptamer nature and selection. The most interesting part includes the description of aptamers based radiopharmaceuticals.

I have some suggestions how this manuscript could be improved for the better understanding.

1. Despite the words: molecular imaging; radiolabel; targeted imaging are listed as a key words, the paper does not contain figures. The reader would expect to find in this paper how all this imaging looks, ideally in comparison with other non-radioactive methods. Adding the figure with the principal sheme of radiolabeling of the aptamer and targeting will make it easier for the reader to understand and attract people. Most of the readers firstly look at the pictures and then read.

We have added a schematic that demonstrates how targeting ligands attach to chelators and which radioisotopes are effective for different purposes

2. It is a good idea to represent the table 1 with advantages and disadvantages, please include also PET and SPECT to this table.

We have added in PET and SPECT to the table

3. Abstract should be more specific. The number and some specifications and applications of radiolabeled aptamers should be included into the abstract.

We have added more detail to the abstract

4. Please make the table with all aptamer based radiopharmaceuticals.

Which should contain: radioisotope, chelator, aptamer, target, application, reference (may be something else)

We have added a table

Why does the introduction starts with Breast cancer. And only this cancer type is described? Why not lung or brain. I think it should be more general.

Line 163-165: Clinical studies have also been performed with 64Cu-DOTA-trastuzumab as a theranostic tool in breast cancer (HER2+ AND HER2-) After 24 hours post injection, the conjugate appeared to have bound to HER2, and PET was able to produce images which demonstrated a good uptake in HER2+ cases.

No information on HER2-

We have amended this section of the paper

Line 164 (HER2+ AND HER2-) – and should be in lowercase

Thank you for picking up this error – we have amended

Lines 200. Will be good to have picture or table with criteria for developing a novel imaging agents.

Also picture or table with all possible targeting ligands suitable for radiolabeling with advantages or disadvantages

We have added a schematic that demonstrates how targeting ligands attach to chelators and which radioisotopes are effective for different purposes

Line 207 aptamers could be up to 100 nucleotides

We have amended this

Line 223 you described selection but looks like as the last  pool will be sequenced, it is better to write that after analyzing binding abilities the pool with the best affinity and selectivity will be sequenced.

We have amended this

Line 223 Now most of researchers prefer NGS sequencing, but not cloning with further sequencing, as it was before.

We agree that most researchers prefer NGS and we have amended this sentence. However, costs still prohibit some researchers from performing NGS on all rounds and may stick to Sanger sequencing.